# Neoadjuvant Concurrent Chemoradiotherapy Versus Neoadjuvant Chemotherapy in Thymic Epithelial Tumors: A Propensity Score-Matched Analysis

**DOI:** 10.3390/cancers18010085

**Published:** 2025-12-27

**Authors:** Bubse Na, Chang Hyun Kang, Taeyoung Yun, Ji Hyeon Park, Kwon Joong Na, Samina Park, Hyun Joo Lee, In Kyu Park, Young Tae Kim, Bhumsuk Keam, Hak Jae Kim

**Affiliations:** 1Department of Thoracic and Cardiovascular Surgery, Seoul National University Hospital, Seoul 03080, Republic of Korea; otlkin87@gmail.com (B.N.);; 2Cancer Research Institute, Seoul National University College of Medicine, Seoul 03080, Republic of Korea; 3Department of Internal Medicine, Seoul National University Hospital, Seoul 03080, Republic of Korea; 4Department of Radiation Oncology, Seoul National University Hospital, Seoul 03080, Republic of Korea

**Keywords:** thymoma, thymic carcinoma, neoadjuvant chemotherapy, neoadjuvant chemoradiotherapy

## Abstract

Thymic epithelial tumors (TETs) are rare malignancies, and evidence guiding the optimal neoadjuvant treatment strategy remains limited. Neoadjuvant chemotherapy (NCT) is commonly used for locally advanced, potentially resectable TETs, while neoadjuvant chemoradiotherapy (NCRT) has been proposed to improve local control. This study is the first to directly compare NCRT and NCT in patients with thymic tumors. Using a retrospective propensity score-matched design, we evaluated surgical and oncologic outcomes between the two strategies. NCRT was associated with significantly higher complete (R0) resection rates and better tumor regression grades, indicating improved local control. However, these advantages did not translate into improved overall or recurrence-free survival. Treatment failure was predominantly driven by regional or distant recurrence rather than local recurrence. These findings suggest that although NCRT may enhance resectability, improving local control alone may be insufficient to improve long-term survival, highlighting the need for more effective systemic treatment strategies in advanced TETs.

## 1. Introduction

Thymic epithelial tumors (TETs) are rare malignancies arising from the anterior mediastinum and include thymoma, thymic carcinoma, and thymic neuroendocrine tumors [1]. Although uncommon, their incidence has been increasing in South Korea and other regions [2]. Advanced TETs frequently present with invasion of adjacent structures or metastatic disease, making complete surgical resection challenging.

For locally advanced but potentially resectable TETs, neoadjuvant chemotherapy (NCT) is commonly recommended to improve resectability and long-term outcomes [3]. Previous studies have reported improved survival and higher complete (R0) resection rates with induction chemotherapy compared with upfront surgery, with reported R0 rates ranging from 69% to 78% [4,5,6,7,8]. To further enhance locoregional control, neoadjuvant chemoradiotherapy (NCRT) has been explored in selected centers, with several single-arm studies suggesting favorable R0 resection rates [9,10,11,12]. However, evidence directly comparing NCRT and NCT is lacking.

Therefore, we conducted a retrospective propensity score–matched analysis to directly compare surgical and oncologic outcomes between NCRT and NCT in patients with advanced TETs, with a particular focus on complete resection rates, pathological response, and long-term survival.

## 2. Materials and Methods

### 2.1. Study Protocol

This study was approved by the Institutional Review Board, and the requirement for informed consent was waived (Approval No.: 2410-058-1576; Approval date: 21 October 2024). Patients who underwent surgery for TETs following NCRT or NCT between January 2009 and June 2022 were retrospectively analyzed. Exclusion criteria included surgery for recurrent TETs, non-curative procedures such as incisional biopsy, rare histological subtypes such as sarcomatoid carcinoma or neuroendocrine carcinoma, and surgery performed more than 6 months after the completion of neoadjuvant treatment (Appendix A). During the study period, a total of 98 patients were included, comprising the NCRT group (*n* = 30) and the NCT group (*n* = 68). The distribution of patients by year of operation is presented (Appendix A).

All patients included in the study underwent a preoperative biopsy prior to receiving neoadjuvant treatment. The choice between NCRT and NCT was made through multidisciplinary discussions, primarily at the discretion of the attending surgeon, based on the potential for complete resection of the primary tumor and the extent of metastatic spread.

NCRT was generally preferred when the tumor was sufficiently localized to be encompassed within the radiation field. In addition, NCRT was more frequently utilized when invasion into critical structures, particularly the great vessels, was suspected. The suspicion of invasion was initially based on patient symptoms and CT findings, and cine MRI was also employed to assess invasion and resectability. As NCT was also employed in groups where NCRT was generally preferred, a propensity score matching study was conducted to derive comparable groups.

NCRT consisted of concurrent chemotherapy and radiotherapy. Radiotherapy was administered at doses of 44–45 Gy in 22 fractions for 22 patients, 50 Gy for 5 patients, and over 60 Gy for 2 patients. In one case, the radiation dose could not be determined due to incomplete records from an outside institution. In the NCRT group, the most commonly used chemotherapy regimen was cisplatin (Cisplan^®^, Dong-A ST, Seoul, Republic of Korea) monotherapy, administered to 25 patients (83.3%). In the NCT group, the most frequently used regimen was a combination of cyclophosphamide (Endoxane^®^, Baxter, Deerfield, IL, USA), doxorubicin (Adriamycin^®^, Pfizer, New York, NY, USA), and cisplatin, used in 57 patients (83.8%).

Demographic, radiologic, and clinicopathologic data were retrieved from electronic medical records. These included age, sex, smoking history, comorbidities (such as hypertension, diabetes mellitus, and myasthenia gravis), histologic type, clinical stage, response to neoadjuvant treatment, surgical approach, presence of combined resection, resection margin status, tumor regression grade (TRG) [13], World Health Organization (WHO) histological classification, pathologic stage, receipt of adjuvant chemotherapy or radiotherapy, in-hospital and 90-day mortality, and the types and grades of postoperative complications [14]. Regarding resection margin, R0 resection was strictly defined as the absence of tumor cells at the inked surgical margin, regardless of the presence of capsular invasion or soft tissue infiltration beyond the capsule. All specimens were reviewed by 3 board-certified thymic pathology-dedicated pathologists.

Clinical and pathological staging were evaluated according to the 8th edition of the TNM staging system [15], as the study period extended through June 2022. The response to neoadjuvant treatment was assessed based on the Response Evaluation Criteria in Solid Tumors version 1.1 (RECIST 1.1) [16]. Tumor size was measured as the largest diameter on axial cross-sectional CT images. A partial response was defined as a minimum 30% decrease in tumor size following 4 weeks of therapy. Progressive disease was defined by an increase in tumor size exceeding 20%, while stable disease referred to cases that did not meet the criteria for either partial response or progressive disease.

### 2.2. Outcome Measurement

The primary outcome was the rate of complete resection (R0). R1 resection was defined as the presence of microscopic tumor cells at the margin of resection specimen, while R2 resection was defined as the presence of gross residual primary mass or metastatic lesions.

The secondary outcomes included early surgical outcomes such as pathological results, postoperative complications, and in-hospital mortality. Overall survival (OS), recurrence-free survival (RFS), and recurrence patterns were also pursued as secondary objectives. Patients were followed every 3 to 6 months for thymic carcinoma and every 6 months for thymoma during the first two years. Thereafter, the follow-up interval was extended to every 6 to12 months. OS was defined as the time from the date of surgery to the date of death or last follow-up. The date of recurrence was defined as the date on which the disease was first clinically suspected. RFS was defined as the time from the date of surgery to the date of the first documented event, either recurrence or death, whichever occurred first.

Recurrence sites or patterns were classified according to the recommendations by the International Thymic Malignancy Interest Group (ITMIG) published in 2010 [17] based on the site of first recurrence. Local recurrences were defined as recurrence in the anterior mediastinum, thymic bed, or immediately adjacent structures. Regional recurrences were defined as pleural or pericardial nodules, mediastinal lymph nodes or intrathoracic lesions not contiguous with the thymus. Distant recurrences were defined as extrathoracic or intraparenchymal pulmonary nodules. When recurrence sites involved multiple lesions, each lesion was recorded under its respective classification, allowing for overlapping entries. In analyses limited to patients with recurrence, the patterns (local, regional, distant) were defined in a mutually exclusive manner based on the most advanced site.

The last follow-up date for survival and recurrence, based on electronic medical records, was 31 October 2024. For patients lost to follow-up, survival status was updated using national population statistics database, with the data available through 31 December 2023.

### 2.3. Statistical Analysis

Continuous variables were summarized as means with standard deviations or medians with interquartile ranges (IQRs) and compared using Student’s *t*-test or Mann–Whitney U test. Categorical variables were reported as counts and percentages. To evaluate prognostic factors for survival, Cox proportional hazards regression was performed. To ensure that no principal variables are omitted from multivariable analysis due to *p*-value-based model building, we conducted univariable analysis on variables based on a priori assumptions on causal inference. Subsequently, all factors analyzed in the univariable analysis were included in the multivariable analysis, with forward selection applied to control for multicollinearity. Results were presented as hazard ratios (HRs) with corresponding confidence intervals (CIs) and *p*-values.

Survival curves were generated using the Kaplan–Meier method and compared between groups using the log-rank test and Cox proportional hazards regression.

Time-to-event outcomes, specifically the cumulative incidence functions for sites and patterns of initial recurrences were estimated using the prodlim package (version 2023.8.28) of the R software (version 4.2.0). In these analyses, competing risks were considered, with death explicitly defined as a competing event. Comparisons of cumulative incidence functions between groups were performed using Gray’s test, and Gray’s *p*-values were used to assess between-group differences while accounting for competing risks. For the recurrence site, patients who experienced specific sites of initial recurrence were regarded as having an event in the analysis. Patients who were lost to follow-up were censored at their last known follow-up time. Outcomes were then compared between the NCRT and NCT groups. For the recurrence pattern, patients who developed the recurrence pattern of interest were regarded as having an event, whereas those who developed other patterns of recurrence were additionally considered censored. Cumulative incidence rate curves of recurrence patterns in pre-matched population were presented.

Propensity score matching was performed to compare the NCRT and NCT groups. The covariates included in the model were age, sex, histologic type (thymoma vs. thymic carcinoma), clinical T stage (1/2 vs. 3/4), and clinical M stage (0 vs. 1). These covariates were chosen to balance potential confounding factors. Age and sex were included as fundamental demographic variables. Histologic type was incorporated due to the known poorer prognosis associated with thymic carcinoma. Clinical T stage was considered as it reflects tumor resectability, particularly regarding major vessel involvement (T3/4). Clinical M stage was included to represent the overall spatial extent of the tumor. Clinical T and M stages were appropriate matching variables because they influenced the likelihood of receiving a specific type of neoadjuvant treatment and were associated with the likelihood of achieving complete resection. One-to-one matching was conducted using the optimal method without replacement.

All statistical analyses were performed using the R Software (version 4.2.0, R Foundation for Statistical Computing) and IBM SPSS ver. 26.0 (IBM Corp., Armonk, NY, USA).

## 3. Results

The median follow-up duration for all patients was 51.7 months (IQR, 35.6–74.8). The median follow-up duration was 40.7 months (IQR, 35.5–56.7) in the NCRT group and 55.4 months (IQR, 35.6–94.0) in the NCT group.

The demographic characteristics of the patients are summarized in Table 1. The median age was 56 years. Male patients accounted for 62.2% of the study population (*n* = 61). Biopsy and histologic diagnosis were performed in all patients prior to neoadjuvant treatment, with 42 patients (42.9%) diagnosed with thymic carcinoma. The median tumor size in pre-treatment CT was 7.3 cm (IQR 6.1–8.5). The distribution of clinical stages was as follows: stage II in 2 patients (2.0%), stage III in 30 patients (30.6%), and stage IV in 66 patients (67.3%). After propensity score matching, demographic characteristics were well balanced between the two groups (Appendix A). The rates of intraoperative CPB/ECMO support, types of surgical approach, and combined organ resections are presented (Appendix A).

Responses to neoadjuvant treatment, pathologic results and postoperative courses are summarized in Table 2. Response evaluation by CT scan after neoadjuvant treatment did not show a significant difference between the groups. The NCRT group demonstrated a trend toward a higher partial response rate (36.7% in the NCRT group vs. 26.7% in the NCT group; *p* = 0.405, Table 2) and greater tumor size reduction (25.7 ± 13.6% in the NCRT group vs. 20.2 ± 15.7% in the NCT group; *p* = 0.149); however, these differences were not statistically significant.

The complete resection rate was significantly higher in the NCRT group (93.3%) compared to the NCT group (73.3%; *p* = 0.038) (Figure 1A). The TRG score was significantly lower in the NCRT group (median [IQR], 3 [3,4]) compared to the NCT group (4 [4]) (*p* = 0.002), indicating a better tumor response (Figure 1B). In the NCRT group, the preoperative radiotherapy dose did not significantly affect the complete resection rate (90.9% for 44–45 Gy (*n* = 22) vs. 100% for higher doses (*n* = 7); *p* = 1.000) or the tumor regression grade (median [IQR], 3 [3,4] for 44–45 Gy vs. 3 [2,3,4] for higher doses; *p* = 0.501).

The downstaging rate from clinical N-positive to pathologic N0 was not significantly different between the two groups (83.3% [5/6] in the NCRT group vs. 56.3% [9/16] in the NCT group; *p* = 0.351). Likewise, the downstaging rate from clinical M-positive to M0 was not significantly different between the two groups (42.9% [3/7] in the NCRT group vs. 20.9% [9/43] in the NCT group; *p* = 0.337).

Adjuvant radiotherapy was administered more frequently in the NCT group (10.0% in the NCRT group vs. 53.3% in the NCT group; *p* < 0.001), likely due to the higher rate of incomplete resection. No significant differences in mortality or complication rates were observed between the two groups (Table 2 and Appendix A). One case requiring bleeding control occurred in the NCT group, and one case of transient ischemic attack was observed in the NCRT group. Both groups had one case of thromboembolic complication. Analysis of postoperative complications in relation to CPB/ECMO use showed a tendency for higher thromboembolism rates in the CPB/ECMO use group (Appendix A).

The 5-year OS rates were 83.1% (CI 70.5–97.8%) in the NCRT group and 74.5% (CI 59.8–92.8%) in the NCT group (NCRT vs. NCT, HR = 0.800; CI 0.261–2.453; *p* = 0.7), with no statistically significant difference. When stratified by histological subtype, neither thymoma nor thymic carcinoma showed a significant difference in OS between the two groups (Figure 2A,B). Similarly, the 5-year RFS rates were 31.2% (CI 16.9–57.4%) in the NCRT group and 33.5% (CI 19.9–56.6%) in the NCT group (NCRT vs. NCT, HR = 1.179; CI 0.622–2.235; *p* = 0.62), also without a significant difference. RFS remained comparable between the two groups within both thymoma and thymic carcinoma subgroups (Figure 2A,B).

In subgroup analyses comparing the NCRT group and the NCT group with adjuvant radiotherapy (NCTRT(+)) (*n* = 16) among matched cohorts, the 5-year OS rates were 83.1% (CI 70.5–97.8%) in the NCRT group and 74.0% (CI 55.0–99.6%) in the NCTRT(+) group (NCRT vs. NCTRT(+), HR = 0.692; CI 0.197–2.435; *p* = 0.57), with no statistically significant difference. Similarly, the 5-year RFS rates were 31.2% (CI 16.9–57.4%) in the NCRT group and 31.2% (CI 15.1–64.6%) in the NCTRT(+) group (NCRT vs. NCTRT(+), HR = 1.022; CI 0.488–2.143; *p* = 0.954), also without a significant difference.

Regarding recurrence sites, regional recurrence was the most common in the pre-matched population, accounting for 52.0%, followed by distant (20.4%) and local recurrence (12.2%) (Table 3). The NCRT group exhibited a numerically lower incidence of local recurrence compared to the NCT group, both in the pre-matched (3.7% vs. 25.7%, *p* = 0.13) and matched populations (3.7% vs. 25.2%, *p* = 0.13), based on the 5-year cumulative incidence derived from competing risk analysis. However, these differences did not reach statistical significance, likely due to the limited sample size.

In pre-matched cohorts, 5-year cumulative incidence rates of recurrence pattern, with death treated as a competing risk, were 10.8%, 53.0%, 23.3% for local, regional, and distant recurrence patterns, respectively (Appendix A).

Long-term survival was closely correlated with the pattern of recurrence. For 5 patients with initial local recurrence (5 thymoma, 0 thymic carcinoma), both the 5-year overall survival and post-recurrence survival rates were 100%. Patients with distant recurrence exhibited the poorest prognosis, with 5-year OS rates of 46.0%, compared to 90.5% for regional and 100% for local recurrence (*p* = 0.030 for local vs. distant; *p* < 0.001 for regional vs. distant) (Figure 3). Consistent trends were observed for 5-year recurrence-free survival (5.0% for distant, 5.0% for regional, and 20% for local recurrence; *p* = 0.024 for local vs. distant; *p* = 0.021 for regional vs. distant) and 5-year post-recurrence survival (19.2%, 81.4%, and 100%, respectively; *p* = 0.023 for local vs. distant; *p* < 0.001 for regional vs. distant).

Cox-regression analysis was performed to identify risk factors associated with OS. In pre-matched cohort, univariable and multivariable analyses identified the following as statistically significant risk factors for OS: ever smoker (HR = 2.843; 95% CI 1.214–6.657; *p* = 0.016), myasthenia gravis (HR = 3.847, 95% CI 1.014–14.590; *p* = 0.048), thymic carcinoma (HR = 3.656, 95% CI 1.406–9.509, *p* = 0.008), and complete resection (HR = 0.301, 95% CI 0.130–0.700, *p* = 0.005) (Appendix A).

Regarding RFS, univariable and multivariable analyses identified the following as statistically significant risk factors for RFS: ever smoker (HR = 2.075; 95% CI 1.235–3.486; *p* = 0.006), pathologic stage IV vs. I + II (HR = 2.179; 95% CI 1.087–4.369; *p* = 0.028), pathologic stage IV vs. III (HR = 2.165, 95% CI 1.195–3.922, *p* = 0.011) and complete resection (HR = 0.528, 95% CI 0.301–0.927, *p* = 0.026) (Appendix A).

Comparing patients with complete resection (*n* = 79) and incomplete resection (*n* = 19), the 5-year OS rates were 84.1% (CI 75.1–94.2%) in the complete resection group and 55.3% (CI 35.9–85.1%) in the incomplete resection group. The hazard ratio for complete resection compared to incomplete resection was 0.306 (CI 0.136–0.688; *p* = 0.004). The 5-year RFS rates were 33.7% (CI 23.9–47.5%) in the complete resection group and 5.3% (CI 0.8–35.5%) in the incomplete resection group. The hazard ratio for complete resection compared to incomplete resection was 0.427 (CI 0.248–0.734; *p* = 0.002). Among the incomplete resection group, the initial recurrence pattern was regional in 10 patients (62.5%), distant in 6 patients (18.8%), and none was restricted to local recurrence. Three patients (15.8%) in the incomplete resection group did not experience recurrence.

## 4. Discussion

This study compared long-term outcomes between NCRT and NCT in patients with advanced TETs. NCRT demonstrated superior local control, evidenced by lower TRG scores and a higher complete resection rate. Given that complete resection has been closely associated with improved OS in previous studies, it was anticipated that enhanced local control would translate into improved OS or RFS. However, no significant differences in OS or RFS were observed between the NCRT and NCT groups in this study, likely because long-term survival was primarily determined by the regional or distant recurrence rather than local recurrence. These findings suggest that improving local control alone may have a limited impact on overall long-term outcomes.

Although TETs, particularly thymoma, are generally indolent with a favorable prognosis, the long-term outcome of the advanced TETs remains poor [18]. Invasion of adjacent organs and a higher incidence of regional or distant metastases often preclude complete surgical resection. Historically, complete resection has been recognized as a key prognostic factor following surgical treatment of TET [19,20,21,22]. Consequently, many studies have focused on induction therapies aimed at improving resectability. Currently, NCT is the most widely used approach. Several studies have demonstrated favorable survival outcomes with NCT in advanced TETs, and NCCN guidelines also recommend it for potentially resectable thymic tumors [23,24,25,26,27].

However, the outcomes of R0 resection rates have varied across studies evaluating NCT, ranging from 69% to 78% [6,7,8]. Moreover, the absolute rates have remained suboptimal. To enhance resectability, several studies have explored the addition of radiotherapy. To date, only two single-arm studies have reported outcomes following NCRT and surgery, with R0 resection rates of 77% (17/21) [10] and 80% (8/10) [9]. Although R0 resection rates appeared improved with NCRT, no direct comparative studies between NCRT and NCT have been reported to date.

From this perspective, a comparison of NCRT and NCT was conducted in this study, focusing on the R0 resection rate. Local control was assessed by comparing clinical response, pathological response, and R0 resection rates. Clinical response, evaluated by changes in tumor size on CT scan, showed a trend toward greater reduction in the NCRT group, although the difference was not statistically significant. In contrast, pathological response evaluation demonstrated that the TRG was significantly lower in the NCRT group than in the NCT group, suggesting superior tumor response with NCRT [27]. Consequently, the R0 resection rate was 93.3% in the NCRT group, significantly higher than 73.3% in the NCT group, with R0 resection rate of the NCT group consistent with previously reported historical data. These findings suggest that NCRT provides superior local tumor control compared to NCT.

Despite the higher complete (R0) resection rate in the NCRT group, this study could not identify a survival benefit of NCRT over NCT.

In this study, 53.3% of patients in the NCT group received adjuvant radiotherapy [NCTRT(+)], which likely contributed to the improved survival outcomes observed in the NCT group [28,29]. However, no apparent differences in survival outcomes were observed when comparing NCRT with NCT overall, or when comparing NCRT with NCTRT(+) alone within our cohort. A meta-analysis reported that the addition of adjuvant radiotherapy in patients who underwent surgery alone with R0 resection improved overall survival but failed to improve disease-free survival [30]. Given that our patient population differed—having already received neoadjuvant chemotherapy and including patients with incomplete resections—the impact of adjuvant radiotherapy might have been less pronounced than expected.

Survival differences were more pronounced when comparing patients with complete (R0) and incomplete resections. In patients with incomplete resections, initial progressions or recurrences predominantly occurred in regional or distant sites rather than at the primary tumor site. Moreover, in the entire cohort, regional and distant recurrences were the major patterns of treatment failure and were associated with significantly poorer prognosis compared to local recurrence. In contrast, the incidence of local recurrence was low, and prognosis following local recurrence was relatively favorable.

These findings highlight the importance of achieving R0 resection regardless of the type of neoadjuvant treatment, as residual disease does not necessarily correlate with local recurrence. Nonetheless, focusing solely on controlling the primary tumor and preventing local recurrence cannot be considered the ultimate goal in the management of advanced thymic epithelial tumors (TET), as this approach alone is insufficient to improve long-term survival. Future strategies aimed at preventing regional and distant recurrence are likely critical to improving long-term survival in patients with advanced TETs.

This study has several limitations. First, it was a retrospective analysis. Although propensity score matching was employed to minimize confounding, minor imbalances in some variables remained. Second, most patients in the NCRT group were treated after 2018, and owing to the disease’s rarity, potential temporal effects related to surgical advancements or institutional changes could not be adjusted for. Third, although patient selection was made through multidisciplinary discussion, NCRT was more commonly administered to patients with more advanced T stages, whereas NCT was more often given to those with regional or distant metastases. Fourth, the chemotherapy backbone differed between groups, with cisplatin monotherapy commonly used in the NCRT group to reduce concurrent radiotherapy–related toxicity, which may have limited the comparability of systemic disease control between NCRT and NCT. Finally, this study is limited by its relatively small sample size, particularly after propensity score-matching, which reduced the number of survival events and limited the power to detect differences in overall survival. Accordingly, survival analyses in this study should be interpreted as exploratory rather than definitive. Nevertheless, these findings provide a rationale and foundation for future studies aimed at clarifying the optimal neoadjuvant strategy in advanced thymic epithelial tumors.

## 5. Conclusions

The NCRT group demonstrated a significantly higher R0 resection rate and better TRG compared to the NCT group. However, despite the improved local control with NCRT, no survival benefit was observed, likely because the predominant pattern of treatment failure was regional or distant recurrences. These findings provide a foundation for further investigations that may ultimately impact the standard of care. More effective neoadjuvant treatment regimens are required to improve the long-term survival outcomes in patients with advanced TETs.

## Figures and Tables

**Figure 1 cancers-18-00085-f001:**
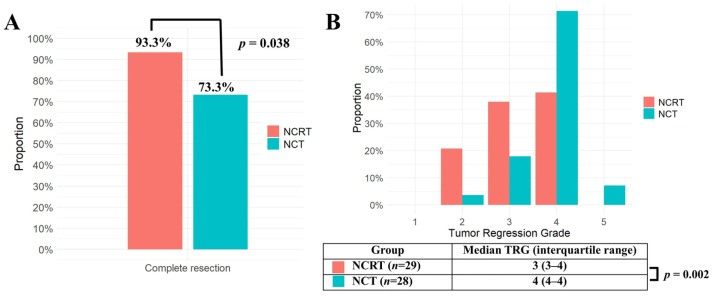
(**A**) Complete resection rates of the NCRT group (*n* = 30) and the NCT group (*n* = 30) in matched population were compared using the Chi-square test. (**B**) Tumor Regression Grades (TRGs) of the NCRT group (*n* = 29) and the NCT group (*n* = 28) in the matched population were compared using Mann–Whitney U test. TRG was missing in 1 patient from the NCRT group and 2 patients from the NCT group. NCRT: neoadjuvant chemoradiotherapy, NCT: neoadjuvant chemotherapy, *p*: probability value, TRG: tumor regression grade.

**Figure 2 cancers-18-00085-f002:**
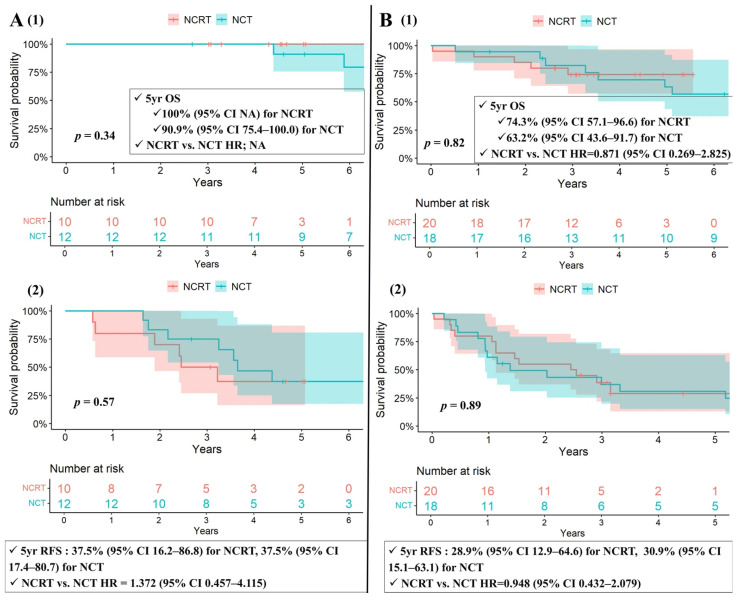
(**A**) Kaplan–Meier survival curves for patients with thymoma (1) OS and (2) RFS among matched cohorts. (**B**) Kaplan–Meier survival curves for patients with thymic carcinoma (1) OS and (2) RFS among matched cohorts. CI: confidence interval, NA: not available, NCRT: neoadjuvant chemoradiotherapy, NCT: neoadjuvant chemotherapy, OS: overall survival, *p*: probability value, RFS: recurrence-free survival, Yr: year.

**Figure 3 cancers-18-00085-f003:**
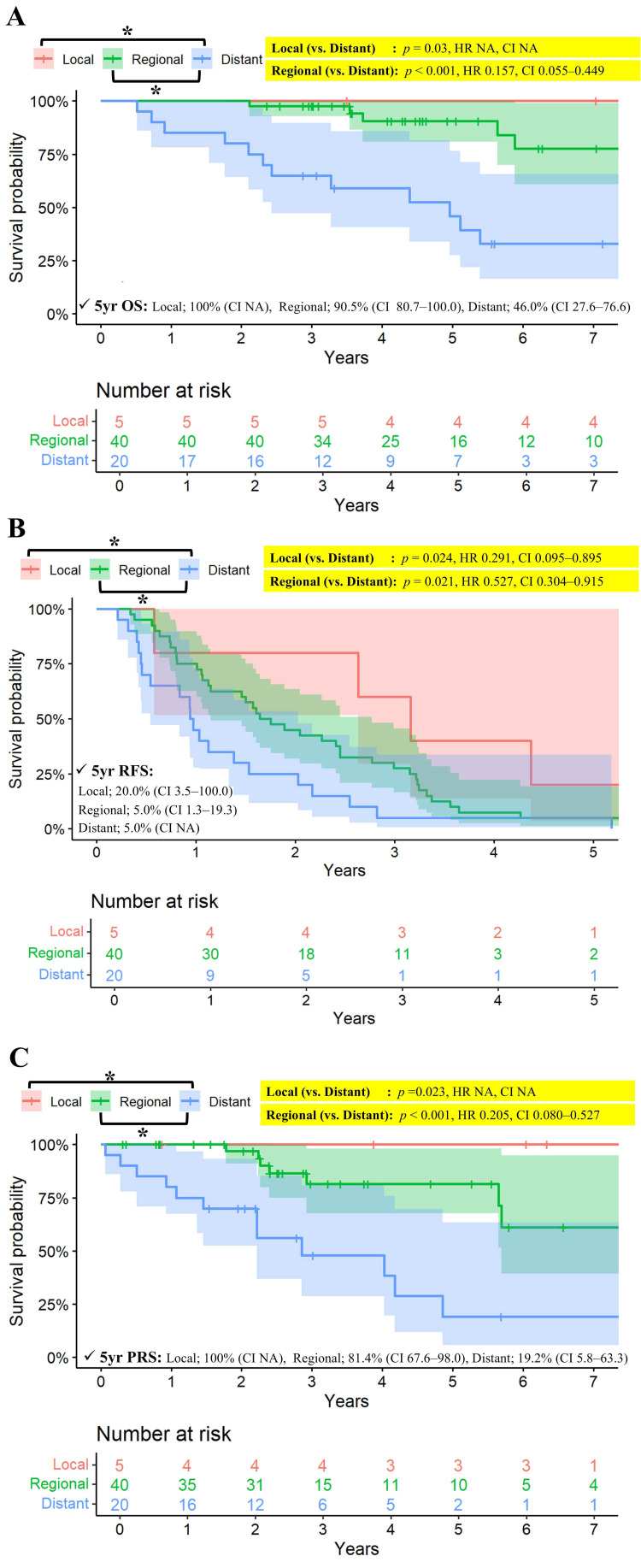
(**A**) Kaplan–Meier survival curves for OS stratified by pattern of recurrence among patients with recurrence. (**B**) Kaplan–Meier curves for FFR stratified by pattern of recurrence among patients with recurrence. (**C**) Kaplan–Meier survival curves for post-recurrence survival stratified by pattern of recurrence among patients with recurrence. CI: confidence interval, FFR: Freedom-from recurrence, NA: not available, OS: overall survival, *p*: probability value, RFS: recurrence-free survival, Yr: year. * Indicates a statistically significant comparison (*p* < 0.05). Since there was a group with no events, the *p*-value was calculated using the log-rank test, while the hazard ratio (HR) and confidence interval (CI) were estimated using Cox regression. All confidence intervals (CI) are reported at the 95% confidence level.

**Table 1 cancers-18-00085-t001:** Demographic data of the patients.

Variable		Pre-Matched Patients			PS-Matched Patients		
	Total (*n* = 98)	NCRT (*n* = 30)	NCT (*n* = 68)	*p*	NCRT (*n* = 30)	NCT (*n* = 30)	*p*
Age	56 (44–63)	56 (44–60)	56 (44–64)	0.728	56 (44–60)	55 (44–64)	0.997
Male	62.2% (*n* = 61)	66.7% (*n* = 20)	60.3% (*n* = 41)	0.549	66.7% (*n* = 20)	56.7% (*n* = 17)	0.426
Smoking history				0.345			0.465
Never smoker	76.5% (*n* = 75)	76.7% (*n* = 23)	76.5% (*n* = 52)		76.7% (*n* = 23)	73.3% (*n* = 22)	
Ex-smoker	15.3% (*n* = 15)	10.0% (*n* = 3)	17.6% (*n* = 12)		10.0% (*n* = 3)	20.0% (*n* = 6)	
Current smoker	8.2% (*n* = 8)	13.3% (*n* = 4)	5.9% (*n* = 4)		13.3% (*n* = 4)	6.7% (*n* = 2)	
HTN	19.4% (*n* = 19)	8.2% (*n* = 4)	22.1% (*n* = 15)	0.314	8.2% (*n* = 4)	30.0% (*n* = 9)	0.117
DM	11.2% (*n* = 11)	3.3% (*n* = 1)	14.7% (*n* = 10)	0.100	3.3% (*n* = 1)	10.0% (*n* = 3)	0.301
Myasthenia gravis	8.2% (*n* = 8)	0% (*n* = 0)	11.8% (*n* = 8)	0.102	0% (*n* = 0)	6.7% (*n* = 2)	0.492
Thymic carcinoma	42.9% (*n* = 42)	66.7% (*n* = 20)	32.4% (*n* = 22)	0.002	66.7% (*n* = 20)	60.0% (*n* = 18)	0.592
Mass size in CT (cm)	7.3 (6.1–8.5)	6.6 (6.2–7.9)	7.5 (6.0–8.9)	0.221	6.6 (6.2–7.9)	7.9 (6.6–8.8)	0.063
cT stage				0.015			0.262
T1/T2	21.4% (*n* = 21)	10.0% (*n* = 3)	26.5% (*n* = 18)		10.0% (*n* = 3)	13.3% (*n* = 4)	
T3	36.7% (*n* = 36)	30.0% (*n* = 9)	39.7% (*n* = 27)		30.0% (*n* = 9)	46.7% (*n* = 14)	
T4	41.8% (*n* = 41)	60.0% (*n* = 18)	33.8% (*n* = 23)		60.0% (*n* = 18)	40.0% (*n* = 12)	
cN stage				0.700			0.371
N0	77.6% (*n* = 76)	80.0% (*n* = 24)	76.5% (*n* = 52)		80.0% (*n* = 24)	70.0% (*n* = 21)	
N1/2	22.4% (*n* = 22)	20.0% (*n* = 6)	23.5% (*n* = 16)		20.0% (*n* = 6)	30.0% (*n* = 9)	
cM stage				<0.001			0.390
M0	46.9% (*n* = 46)	76.7% (*n* = 23)	33.8% (*n* = 23)		76.7% (*n* = 23)	66.7% (*n* = 20)	
M1	53.1% (*n* = 52)	23.3% (*n* = 7)	66.2% (*n* = 45)		23.3% (*n* = 7)	33.3% (*n* = 10)	
cS tage				<0.001			0.088
II	2.0% (*n* = 2)	3.3% (*n* = 1)	1.5% (*n* = 1)		3.3% (*n* = 1)	0% (*n* = 0)	
III	30.6% (*n* = 30)	60.0% (*n* = 18)	17.6% (*n* = 12)		60.0% (*n* = 18)	40.0% (*n* = 12)	
IV	67.3% (*n* = 66)	36.7% (*n* = 11)	80.9% (*n* = 55)		36.7% (*n* = 11)	60.0% (*n* = 18)	

c, clinical; CT, computed tomography; DM, diabetes mellitus; HTN, hypertension; NCRT, neoadjuvant chemoradiotherapy; NCT, neoadjuvant chemotherapy; PS, propensity score.

**Table 2 cancers-18-00085-t002:** Response to neoadjuvant treatment, pathologic results and postoperative courses.

Variable		Pre-Matched Patients			PS-Matched Patients		
	Total (*n* = 98)	NCRT (*n* = 30)	NCT (*n* = 68)	*p*	NCRT (*n* = 30)	NCT *(n* = 30)	*p*
Response to nTx				0.239			0.405
PR	28.6% (*n* = 28)	36.7% (*n* = 11)	25.0% (*n* = 17)		36.7% (*n* = 11)	26.7% (*n* = 8)	
SD	71.4% (*n* = 70)	63.3% (*n* = 19)	75.0% (*n* = 51)		63.3% (*n* = 19)	73.3% (*n* = 22)	
Mass size reduction (%) in CT after nTx	22.6 ± 14.6	25.7 ± 13.6	21.3 ± 14.9	0.166	25.7 ± 13.6	20.2 ± 15.7	0.149
Complete resection	80.6% (*n* = 79)	93.3% (*n* = 28)	75.0% (*n* = 51)	0.034	93.3% (*n* = 28)	73.3% (*n* = 22)	0.038
TRG *^, #^ (*n* = 90)				<0.001			0.002
Grade 2	7.8% (*n* = 7)	20.7% (*n* = 6)	1.6% (*n* = 1)		20.7% (*n* = 6)	3.6% (*n* = 1)	
Grade 3	20.0% (*n* = 18)	37.9% (*n* = 11)	11.5% (*n* = 7)		37.9% (*n* = 11)	17.9% (*n* = 5)	
Grade 4	64.4% (*n* = 58)	41.4% (*n* = 12)	75.4% (*n* = 46)		41.4% (*n* = 12)	71.4% (*n* = 20)	
Grade 5	7.8% (*n* = 7)	0% (*n* = 0)	11.5% (*n* = 7)		0% (*n* = 0)	7.1% (*n* = 2)	
Necrosis^#^(%) (*n* = 80)	11.6 ± 20.6 (*n* = 80)	17.8 ± 30.1 (*n* = 22)	9.3 ± 15.3 (*n* = 58)	0.214	17.8 ± 30.1 (*n* = 22)	11.9 ± 18.3 (*n* = 27)	0.394
WHO type B2/B3 among thymoma (*n* = 56)	76.8% (*n* = 43)	70% (*n* = 7)	78.3% (*n* = 36)	0.575	0% (*n* = 0)	75.0% (*n* = 9)	1
ypT stage				0.330			0.319
T1/T2	30.6% (*n* = 30)	40.0% (*n* = 12)	26.5% (*n* = 18)		40.0% (*n* = 12)	30.0% (*n* = 9)	
T3	58.2% (*n* = 57)	53.3% (*n* = 16)	60.3% (*n* = 41)		53.3% (*n* = 16)	53.3% (*n* = 16)	
T4	11.2% (*n* = 11)	6.7% (*n* = 2)	13.2% (*n* = 9)		6.7% (*n* = 2)	16.7% (*n* = 5)	
ypN+	14.3% (*n* = 14)	6.7% (*n* = 2)	17.6% (*n* = 12)	0.215	6.7% (*n* = 2)	26.7% (*n* = 8)	0.080
ypM1	48.0% (*n* = 47)	13.3% (*n* = 4)	63.2% (*n* = 43)	<0.001	13.3% (*n* = 4)	46.7% (*n* = 14)	0.005
ypStage				<0.001			0.016
I	13.3% (*n* = 13)	23.3% (*n* = 7)	8.8% (*n* = 6)		23.3% (*n* = 7)	10.0% (*n* = 3)	
II	7.1% (*n* = 7)	13.3% (*n* = 4)	4.4% (*n* = 3)		13.3% (*n* = 4)	6.7% (*n* = 2)	
III	26.5% (*n* = 26)	43.3% (*n* = 13)	19.1% (*n* = 13)		43.3% (*n* = 13)	26.7% (*n* = 8)	
IV	53.1% (*n* = 52)	20.0% (*n* = 6)	67.6% (*n* = 46)		20.0% (*n* = 6)	56.7% (*n* = 17)	
Adjuvant CTx	17.3% (*n* = 17)	20.0% (*n* = 6)	16.2% (*n* = 11)	0.773	20.0% (*n* = 6)	20.0% (*n* = 6)	1.000
Adjuvant RTx	30.6% (*n* = 30)	10.0% (*n* = 3)	39.7% (*n* = 27)	0.004	10.0% (*n* = 3)	53.3% (*n* = 16)	<0.001
In-hospital mortality	2.0% (*n* = 2)	3.3% (*n* = 1)	1.5% (*n* = 1)	0.521	3.3% (*n* = 1)	0% (*n* = 0)	1.000
90-day mortality	1.0% (*n* = 1)	3.3% (*n* = 1)	0% (*n* = 0)	0.306	3.3% (*n* = 1)	0% (*n* = 0)	1.000

CT, computed tomography; CTx, chemotherapy; NCRT, neoadjuvant chemoradiotherapy; NCT, neoadjuvant chemotherapy; nTx, neoadjuvant treatment; PR, partial response; PS, propensity score; RTx, radiotherapy; SD, stable disease; TRG, tumor regression grade; WHO, World Health Organization. * Grade 1: Complete pathologic response, no viable tumor, Grade 2: Rare residual viable tumor cells scattered through fibrosis, Grade 3: Increase in number of residual tumor cells; fibrosis predominant, Grade 4: Residual viable tumor outgrowing fibrosis, Grade 5: No regressive changes; ^#^ For variables with missing values, the number of available cases (n) is reported.

**Table 3 cancers-18-00085-t003:** Recurrence sites and patterns in pre-matched patients and matched patients, with 5-year cumulative incidence with death defined as a competing risk.

**All Patients**							
**Recurrence Sites**		**Pre-Matched Patients**			**PS-Matched Patients**		
	**Total** **(*n* = 98)**	**NCRT** **(*n* = 30)**	**NCT** **(*n* = 68)**	**Gray’s *p* Value**	**NCRT** **(*n* = 30)**	**NCT** **(*n* = 30)**	**Gray’s *p* Value**
Local	12.2% (*n* = 12)	3.3% (*n* = 1)	16.2% (*n* = 11)		3.3% (*n* = 1)	16.7% (*n* = 5)	
5-year cumulative incidence (95% CI)	19.9% (7.7–32.1%)	3.7% (0.0–11.0%)	25.7% (10.0–41.4%)	0.13	3.7% (0.0–11.0%)	25.2% (3.6–46.8%)	0.13
Regional	52.0% (*n* = 51)	43.3% (*n* = 13)	55.9% (*n* = 38)		43.3% (*n* = 13)	56.7% (*n* = 17)	
5-year cumulative incidence (95% CI)	58.8% (47.3–70.3%)	55.0% (31.7–78.3%)	60.2% (46.7–73.7%)	0.67	55.0% (31.7–78.3%)	57.5% (37.4–77.6%)	0.93
Distant	20.4% (*n* = 20)	16.7% (*n* = 5)	22.1% (*n* = 15)		16.7% (*n* = 5)	33.3% (*n* = 10)	
5-year cumulative incidence (95% CI)	23.3% (13.8–32.8%)	20.4% (1.4–39.4%)	24.7% (16.1–33.3%)	0.61	20.4% (1.4–39.4%)	32.6% (14.4–50.8%)	0.26
**Only Patients with Recurrence**							
**Recurrence Pattern**	**Total** **(*n* = 65)**		**NCRT** **(*n* = 17)**		**NCT** **(*n* = 48)**		***p* Value**
Local	7.7% (*n* = 5)		5.9% (*n* = 1)		8.3% (*n* = 4)		1.000
Regional	61.5% (*n* = 40)		64.7% (*n* = 11)		60.4% (*n* = 29)		0.755
Distant	30.8% (*n* = 20)		29.4% (*n* = 5)		31.3% (*n* = 15)		0.888

CI, confidence interval; NCRT, neoadjuvant chemoradiotherapy; NCT, neoadjuvant chemotherapy; PS, propensity score.

## Data Availability

Data available on request due to restrictions.

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
