# Peer review of "Neoadjuvant Concurrent Chemoradiotherapy Versus Neoadjuvant Chemotherapy in Thymic Epithelial Tumors: A Propensity Score-Matched Analysis"

_cancers, 2025, doi:10.3390/cancers18010085_

Round 1
Reviewer 1 Report
Comments and Suggestions for Authors
This is a clearly written manuscript summarizing a clinical study comparing induction chemoradiotherapy and chemotherapy in thymic tumors. Advantages are reported for NCRT vs NCT in R0 and tumor regression but overall survival was not significantly different and the authors consider that further investigation with longer follow-up and more patients will be needed to establish therapeutic benefit in survival. At present the study is of limited significance but provides a foundation for further investigation that could impact standard of care. There are a few typographical errors and the text in the figures is too small to read easily, particularly Figure 1B. I did not note a definition of Gray’s p-value used in Table 3 and this should be explained in the statistical analysis.
Author Response
Please see the attachment.
|
Response to Reviewer 1 Comments
|
||
|
1. Summary |
|
|
|
Thank you very much for taking the time to review this manuscript. Please find the detailed responses below and the corresponding revisions/corrections highlighted/in track changes in the re-submitted files.
|
||
|
2. Questions for General Evaluation |
Reviewer’s Evaluation |
Response and Revisions |
|
Does the introduction provide sufficient background and include all relevant references? |
Yes |
|
|
Is the research design appropriate? |
Yes |
|
|
Are the methods adequately described? |
Yes |
|
|
Are the results clearly presented? |
Yes |
|
|
Are the conclusions supported by the results? |
Can be improved |
We appreciate the reviewer’s thoughtful comment. We concur that this study will be fundamental for future investigations that may influence the standard of care as noted below, and we have incorporated this point into the Conclusion section. |
|
Are all figures and tables clear and well-presented? |
Must be improved |
We appreciate the reviewer’s thoughtful comment. We have revised the figure text to improve readability, as noted in our point-by-point response below. |
|
3. Point-by-point response to Comments and Suggestions for Authors |
||
|
Comments 1: This is a clearly written manuscript summarizing a clinical study comparing induction chemoradiotherapy and chemotherapy in thymic tumors. Advantages are reported for NCRT vs NCT in R0 and tumor regression but overall survival was not significantly different and the authors consider that further investigation with longer follow-up and more patients will be needed to establish therapeutic benefit in survival. At present the study is of limited significance but provides a foundation for further investigation that could impact standard of care. There are a few typographical errors and the text in the figures is too small to read easily, particularly Figure 1B. I did not note a definition of Gray’s p-value used in Table 3 and this should be explained in the statistical analysis.
|
||
|
Response 1: We appreciate the reviewer’s thoughtful comment. We concur with your point and added in conclusion that this study may be a foundation for future research. We corrected a typographical error and revised the figure sizes and text sizes to improve readability. We also added a definition of Gray’s p-value in the Statistical Analysis section.
(Conclusion, Page 14, Line 417~418) These findings provide a foundation for further investigations that may ultimately impact the standard of care.
(Table 2, Page 7, Line 216) In-hospital mortality
|
||
|
(Figure 1) (Figure 2)
(Materials and Methods, Page 4, Line 167~169) Comparisons of cumulative incidence functions between groups were performed using Gray’s test, and Gray’s p-values were used to assess between-group differences while accounting for competing risks.
|
||
|
|
||
|
4. Response to Comments on the Quality of English Language |
||
|
Point 1: none |
||
|
|
||
|
5. Additional clarifications none |
||
|
|
||

Reviewer 2 Report
Comments and Suggestions for Authors
This retrospective study compares neoadjuvant chemoradiotherapy (NCRT) versus neoadjuvant chemotherapy (NCT) in thymic epithelial tumors (TETs). It shows that NCRT improves complete resection (R0) rates and tumor regression grade but does not translate into a survival advantage. This is an important topic because evidence guiding induction therapy in TETs is scarce.
1.Although PSM is performed, substantial baseline differences remain pre-matching. The rationale for selecting matching variables is correct but incomplete. Important clinical variables (tumor size, MG status, biopsy-proven carcinoma vs thymoma proportion, suspected vessel invasion) were not included.
2. The manuscript states that regional and distant recurrence dominate prognosis, but it does not analyze why NCRT fails to affect these patterns. Did NCRT reduce nodal or pleural metastasis?
3. Chemotherapy backbone differs dramatically: NCRT: mostly cisplatin monotherapy; NCT: mostly CAP (cyclophosphamide–doxorubicin–cisplatin). This is not a fair comparison in systemic control.
4. Sample size small (n=30 per group after PSM); OS difference is visually present but statistically insignificant due to low events.
5. Since R0 is the primary endpoint, margin assessment methodology must be standardized. Did multiple pathologists review? Was “capsular invasion vs soft tissue infiltration” consistently handled?
6. Did RT dose variability (44–60 Gy) affect TRG or R0? (no subgroup analysis).
Comments on the Quality of English Language1. Introduction is overly long; reduce guideline summary and highlight research gap earlier.
2. Methods section could benefit from a diagram of patient flow.
Round 2
Reviewer 1 Report
Comments and Suggestions for Authors
The revised manuscript addresses my prior concerns and is acceptable for publication